# Variability of autonomic nerve activity in dry eye with decreased tear stability

**Minako Kaido** [1,2]*, **Reiko Arita**[2,3], **Yasue Mitsukura**[4], **Reiko Ishida**[5], **Kazuo Tsubota**[2]

**1** Wada Eye Clinic, Chiba, Japan, **2** Tsubota Laboratory, Inc., Tokyo, Japan, **3** Itoh Clinic, Saitama, Japan, **4** Faculty of Science and Technology, Keio University, Kanagawa, Japan, **5** Ishida Eye Clinic, Shizuoka, Japan

* tomoulton777@ff.em-net.ne.jp

**Data Availability Statement:** All relevant data are within the paper and its Supporting Information files.

**Funding:** The authors received no specific funding for this work.

## Abstract

The autonomic nervous system plays a crucial role in the maintenance of homeostasis. Neurogenic inflammation due to decreased stability of tear film may induce changes in autonomic nerve activity, which could be associated with symptom expression. This study aimed to measure biological parameters that represent autonomic nerve activity in dry eye (DE)s caused by tear film dysfunction and investigate their relationship with symptom intensity. This prospective, cross-sectional, comparative study evaluated 34 eyes of 34 participants (mean age: 52.5 ± 13.4 years; range: 20–81 years) without keratoconjunctival damage. Nineteen eyes in the DE group showed DE symptoms and tear break-up time (TBUT) of ≤5 seconds (short TBUT DE); the 15 eyes in the non-DE group showed no DE symptoms. Autonomic nerve activity was measured for 10 minutes—starting and ending 5 minutes before and after instilling ophthalmic solution—and evaluated using the low-frequency component (LF) to the high-frequency component (HF) ratio of heart rate variability (autonomic balance). The pre-ophthalmic solution administration LF/HF ratio was not significantly different ($P = 0.59$) between the two groups, however, the standard deviation of the LF/HF ratio (LF/HF-SD) tended to be higher in the DE group than that in the non-DE group ($P = 0.086$). The DE symptom intensity was significantly related to LF/HF-SD ($P = 0.005$), which significantly decreased after ophthalmic solution administration in the DE group ($P = 0.04$). The large fluctuations in autonomic balance may be key for the understanding of the mechanism underlying DE symptoms.

## Introduction

There is a discrepancy between the symptoms and clinical findings of dry eye (DE) [1,2]. DE is classified into aqueous-deficient DE and evaporative DE [3,4]. However, the concept that decreased tear film stability is of essence to DE is gaining prominence, especially in Asia [5]. Tear film stability is determined by several factors, including tear volume, tear quality, keratoconjunctival epithelial condition, and eyelid condition. DE with decreased tear film stability and symptoms, with or without increased tear evaporation rate, was described as DE with short tear film break-up time (TBUT) in a Japanese epidemiological study [6]. This type of DE

**Competing interests:** The authors have declared that no competing interests exist.

accounts for 80% of all DE cases in Japan; however, its pathogenesis remains largely unknown [6]. The symptoms of DE include eye pain and eye discomfort, but there is no correlation between the severity of symptoms and clinical test findings in DE [1,2]. This discrepancy is especially evident in short TBUT DE [7–9], which has relatively severe symptoms despite having few corneal epithelium clinical test findings [6–9]. As a result, short TBUT DE is integral in discussions concerning DE symptoms.

The Dry Eye Workshop II identified neuropathy as one of the etiologies of DE disease [10]. Neuropathic pain due to neurogenic inflammation caused by damage or irritation to sensory nerve endings on the ocular surface has been implicated as an important cause of DE symptoms [11–14]. Several studies on corneal sensory nerves related to neuropathic pain have focused on transient receptor potential vanilloid 1 (TRPV1), which is associated with pain sensation, and transient receptor potential melastatin 8 (TRPM8), which is associated with temperature change sensation on the corneal surface [15]. In rodent DE models, Bereiter et al. [16] observed high activity in TRPV1, while Kovács et al. [17] observed high activity in TRPM8. In addition, Masuoka et al. [18] showed that both TRPM8 and TRPV1 are expressed in the cold-sensitive corneal nerve endings of individuals with DE and that TRPV1 activation is involved in pain perception. These findings suggest that the aberrant activity of peripheral neurosensitivity may be linked to DE symptoms.

Additionally, higher brain functions may be involved in the onset and progression of DE symptoms [19–21]. We previously used menthol to investigate cold sensitivity and its association with short TBUT DE and found no difference in cold sensitivity between individuals with short TBUT DE and normal eyes. However, there was a difference in whether the cold stimulus was perceived as pleasant or unpleasant [22]; that is, the qualitative perception of cold stimuli in patients with DE may be different from that of patients with normal eyes, while the quantitative perception of cold stimuli may be the same in both DE and normal patients.

The autonomic nervous system plays a crucial role in the maintenance of homeostasis. In the field of DE, the parasympathetic nervous system, which is activated by abdominal breathing, is important to increases tear volume [23]. According to an animal study on lacrimal innervation, the parasympathetic nervous system is more important than the sympathetic nervous system for tear secretion [24]. Neurogenic inflammation due to decreased stability of tear film may induce changes in autonomic nerve activity, associated with symptom expression. Since parasympathetic activity is relatively reduced under stress, it might manifest as discomfort observed in short TBUT DE. Measurement of autonomic nerve activity may be a useful index for objective evaluation of unpleasant emotions associated with DE. In this study, we aimed to investigate the characteristics of autonomic nerve activity and its relationship with symptom intensity in short TBUT DE.

## Materials and methods

The study protocol was reviewed and approved by the Ethics Committee of the Institutional Review Board of Ito Clinic, Saitama, Japan (registration number IRIN2021-11). All procedures were performed according to the ethical standards of the responsible committee on human experimentation (institutional and national) and the Helsinki Declaration of 1964, as revised in 2013. Informed consent was obtained from all participants. This study was registered in the University Hospital Medical Information Network (registration number UMIN000045019).

### Participants

This prospective, cross-sectional, comparative clinical study evaluated the eyes of keratoconjunctival damage-free volunteers (vital staining score <3) at Wada Eye Clinic. Participants

with DE symptoms and TBUTs of ≤5 seconds (short TBUT DE) were designated as the DE group, while those without DE symptoms were designated as the non-DE group (the TBUT value was not considered a criterion of non-DE) by an ophthalmologist. The Schirmer test values were not considered for the diagnosis of short TBUT DE. Exclusion criteria were previous ocular surgery and/or trauma within the last 12 months; anatomic abnormalities in the cornea, conjunctivitis, and/or eyelids; glaucoma; current contact lens use; and systemic diseases that may affect autonomic nerve activity such as heart disease excluding hypertension, diabetes mellitus, Parkinson's disease, pulmonary disease, and Sjögren's syndrome.

### Sample size determination

The sample size was determined based on a previous study [25]. With a mean difference of 0.51 between the two groups, a standard deviation of 0.5, an alpha error of 0.05, and a power of 0.8, the required sample size was 16 in each group. Forty cases were registered for consideration as ineligible cases.

### DE questionnaire

A Japanese version of the Ocular Surface Disease Index (J-OSDI) [26] questionnaire was completed by participants. The J-OSDI, which was translated and culturally adapted from the Ocular Surface Disease Index into Japanese, has good internal consistency, test-retest reliability, and discriminant validity by known-group (DE vs. non-DE) comparisons. J-OSDI scores of ≥13 points were regarded as positive for DE symptoms and those of <13 points as negative.

### Stress check questionnaire

Participants were then given the Brief Job Stress Questionnaire (BJSQ) as a second questionnaire, which includes questions concerning participants' health during the past month [27]. The BJQS consists of 29 questions used to asses 6 parameters (liveliness, frustration, fatigue, anxiety, depression, and physical complaints). Participants choose one of four responses: "almost never," "sometimes," "often," and "always." Scores were then calculated using a conversion table, with participants achieving a score of 6–30 points. The lower the score, the higher the stress the individual is experiencing; the higher the score, the lower the stress. A score of ≤17 points was considered to be indicative of in mental and physical health distress in everyday life.

### Evaluation of the tear function and ocular surface

We performed ocular surface examinations, including TBUT measurement, keratoconjunctival vital staining, and the Schirmer test, before measuring autonomic nerve activity. The TBUT was measured after administering 2 μL preservative-free 1% sodium fluorescein into the conjunctival sac using a micropipette. Keratoconjunctival epithelial staining was evaluated after TBUT measurement. Overall epithelial damage was scored on a scale of 0–9 points, as described previously [28]. The Schirmer test was performed last during the eye surface evaluations.

### Autonomic nerve activity measurement

The autonomic nerve activity was measured > 10 minutes after performing the Schirmer test using the Silmee™ Bar type Lite (TDK, Tokyo, Japan). This biosensor automatically calculates heartbeat intervals, pulse wave intervals, and autonomic nerve activity by measuring and analyzing electrocardiogram pulse wave, acceleration, and skin temperature. Only data for heart

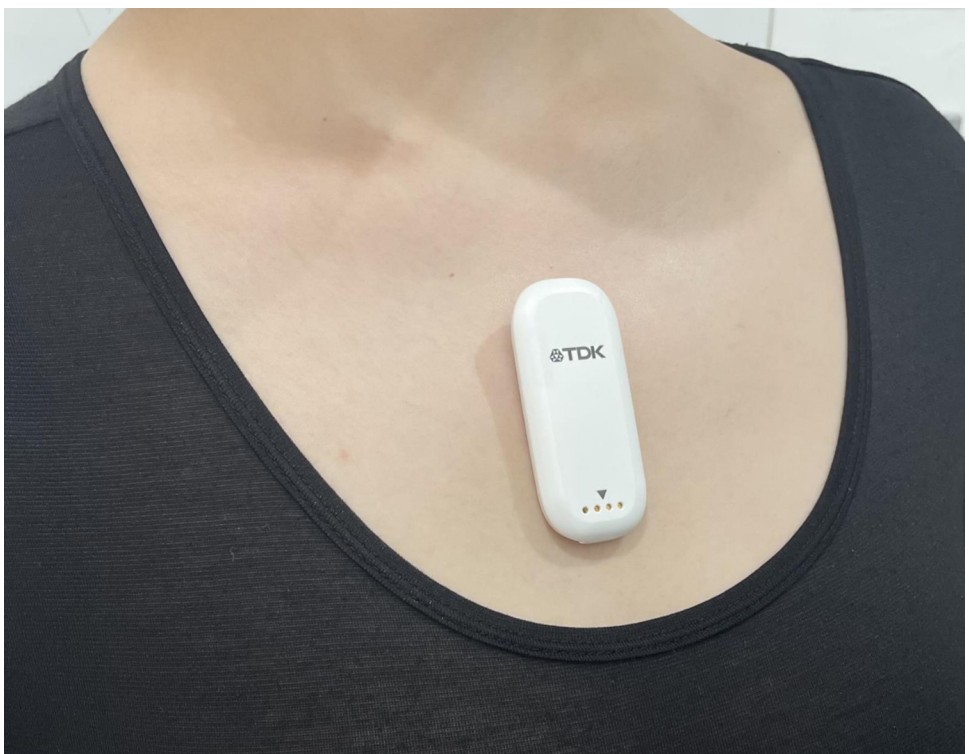

**Fig 1. The Silmee™ Bar type Lite (TDK, Tokyo, Japan) device.** This biosensor automatically calculates heartbeat intervals, pulse wave intervals, and autonomic nerve activity by measuring and analyzing electrocardiograms, pulse waves, acceleration, and/or skin temperature.

rate, RR interval, low-frequency component (LF), high-frequency component (HF), and the LF/HF ratio as autonomic parameters were extracted to avoid excessive amounts of data.

The device was attached approximately 3 cm below the center of both clavicles, a position where the electrocardiographic potential can be measured at a position close to the heart, and the influence of the movement of the upper arm and chest muscles was limited (Fig 1). Autonomic nerve activity was measured with the participant in a rested sitting position and blinking naturally for 10 minutes. We instructed the participants not to sleep, to sit quietly, and to stare blankly into the distance. Measurement was started 5 minutes before ophthalmic solution administration of 0.3% hyaluronic acid (0.1% Hyalein®; Santen Pharmaceutical Co., Ltd., Osaka, Japan) and ended 5 minutes after ophthalmic solution administration. The temperature of the room was maintained at 23˚C –25˚C during examinations, with 50%–60% humidity.

Autonomic nerve function was assessed by frequency analysis of cardiac beat movements (fluctuations in heart rate [RR] intervals) and quantification of sympathetic and parasympathetic activities [29,30]. This technique partitions the total variance (the "power") of a continuous series of beats into its frequency components, typically identifying two main peaks: low frequency (0.04–0.15 Hz) and high frequency (0.15–0.4 Hz) [31]. The HF peak is widely believed to reflect cardiac parasympathetic nerve activity, while the LF is often assumed to have a dominant sympathetic component [29,30,32]. Based on these assumptions, it was proposed that the LF/HF ratio could be used to quantify the degree of sympathovagal balance [29–31,33,34], Heart rate variability metrics obtained in this study were RR, heart rate interval (RR intervals), and the LF/HF ratio.

## Temperature sensation and the comfort level to ophthalmic solution

We evaluated the participants' temperature sensation and comfort level to the ophthalmic solution at room temperature. The temperature sensation and comfort level perceived immediately after administration of the ophthalmic solution were rated using a visual analog scale: a temperature sensation with a score of −5 indicated "most intense cooling sensation imaginable," and a score of +5 indicated "most intense warming sensation imaginable"; a comfort level with a score of −5 indicated "most intense unpleasant sensation imaginable," and a score of +5 indicated "most intense pleasant sensation imaginable."

## Statistical analyses

Data are presented as mean ± standard deviation, where applicable. Baseline J-OSDI scores, tear function, and autonomic nerve activity were compared between both groups using the Mann-Whitney test, since it was difficult to regard the normal distribution as a result of the normality test. Heart rate variability parameters pre- and post-ophthalmic solution administration were compared in each group using the Wilcoxon test. The temperature sensation and the comfort level to the ophthalmic solution were also compared between both groups using the Mann-Whitney test. Multiple regression analysis was performed to determine factors affecting J-OSDI scores, with the pre-ophthalmic solution administration parameters of DE and heart rate variability in the overall cohort as the dependent variables.

All statistical analyses were performed using SPSS Statistics for Windows, version 17.0, (SPSS Inc., Chicago, IL, USA). Statistical significance was set at a $P$-value of $<0.05$.

## Results

### Demographics of the study population

Of the 40 eligible eyes of 40 volunteers with no keratoconjunctival damage (vital staining score $<3$), 34 eyes of 34 participants (11 males, 23 females; mean age: 52.5 ± 13.4 years; range: 20–81 years) were included in this study (S1 Table). There were six participants with DE symptoms and TBUTs of $>5$ seconds who were excluded from the study. Table 1 presents the demographic characteristics of the study participants. The DE and non-DE groups comprised 19 eyes of 19 participants (4 males, 15 females; mean age: 51.4 ± 12.5 years; age range: 20–74

**Table 1. Demographics of the study population.**

|  | DE group n = 19 | Non-DE group n = 15 | P-value* |
|---|---|---|---|
| Age (years) | 51.4 ± 12.5 | 53.9 ± 14.7 | 0.49 |
| Age range (years) | 20–74 | 21–81 | - |
| Male/Female | 4/15 | 7/8 | 0.15 |
| TBUT (seconds) | 3.0 ± 0.9* | 4.6 ± 1.8 | 0.01 |
| VS score (points) | 0.4 ± 0.7 | 0.1 ± 0.5 | 0.17 |
| Schirmer test value (mm) | 4.7 ± 2.6 | 10.2 ± 9.7 | 0.05 |
| J-OSDI (points) | 26.6 ± 10.6 | 6.2 ± 5.0 | <0.001 |
| Mental stress | 20.3 ± 4.6 | 20.9 ± 3.1 | 0.75 |
| Temperature sensitivity to the ophthalmic solution | -1.66 ± 1.06 | -1.32 ± 1.04 | 0.42 |
| Comfort level to ophthalmic solution | 1.8 ± 2.3 | 0.7 ± 1.5 | 0.50 |

*Mann-Whitney test: For comparisons between DE and non-DE groups.

DE, dry eye; TBUT, tear break-up time; VS, vital staining; J-OSDI, Japanese version of the Ocular Surface Disease Index.

years) and 15 eyes of 15 participants (7 males, 8 females; mean age: 53.9 ± 14.7 years; age range: 20–84 years), respectively. Regarding the presence or absence of systemic disease, hypertension was observed in 3 participants each in the DE and non-DE groups, and breast cancer in remission was seen in 1 particpant in the DE group.

The TBUT value was significantly lower ($P < 0.05$), Schirmer test value tended to be lower ($P = 0.05$), and the J-OSDI scores were significantly higher in the DE group than those in the non-DE group ($P < 0.05$). There were no significant differences in the keratoconjunctival staining scores between the two groups ($P > 0.05$).

### Temperature sensation and the comfort level to ophthalmic solution

There were no significant differences in temperature sensitivity to the ophthalmic solution at room temperature (cold stimulus) or stress levels between the DE and non-DE groups ($P > 0.05$; Table 1).

### Autonomic nerve activity

Table 2 shows the results of each heart rate variability metric in the DE and non-DE groups. No significant differences were observed in the pre-ophthalmic solution administration LF, HF, and LF/HF ratio between the DE and non-DE groups ($P > 0.05$). There were no significant changes in the LF, HF, and LF/HF ratio pre- and post-ophthalmic solution administration in the DE group ($P > 0.05$; Fig 2A). Similar results were obtained in the non-DE group ($P > 0.05$).

The pre-ophthalmic solution administration standard deviation of the LF/HF ratio (LF/HF-SD), which represents the fluctuation in the LH/FH ratio over time, tended to be higher in the DE group than that in the non-DE group ($P = 0.086$). It significantly decreased post-ophthalmic solution administration in the DE group ($P = 0.04$), whereas it increased in the non-DE group ($P = 0.05$; Fig 2B). Fig 2C shows the fluctuation of the LF/HF ratio over time in typical DE and non-DE cases.

### Relationship between autonomic nerve activity and DE symptom intensity

Table 3 shows the results of multiple regression analysis for determining factors affecting the J-OSDI scores with independent variables, such as the Schirmer test values, TBUTs,

**Table 2. Heart rate variability.**

|  | Pre-ophthalmic solution administration | | | Post-ophthalmic solution administration | | |
|---|---|---|---|---|---|---|
|  | DE group | Non-DE group | P-value[†] | DE group | Non-DE group | P-value[†] |
| Heart rate | 74.1 ± 11.5 | 72.8 ± 9.7 | 0.54 | 74.2 ± 12.4 | 72.8 ± 9.8 | 0.54 |
| RR interval | 843.8 ± 141.4 | 849.6 ± 114.7 | 0.59 | 840.8 ± 150.3 | 845.4 ± 112.9 | 0.53 |
| LF | 39.9 ± 42.2 | 34.9 ± 19.8 | 0.59 | 34.2 ± 24.3 | 37.7 ± 32.9 | 0.96 |
| LF-SD | 21.9 ± 24.0 | 20.3 ± 20.5 | 0.82 | 17.8 ± 17.3 | 35.1 ± 68.8 | 0.82 |
| HF | 23.6 ± 21.5 | 19.3 ± 15.4 | 0.54 | 20.1 ± 29.3 | 19.8 ± 14.2 | 0.54 |
| HF-SD | 12.8 ± 15.0 | 6.8 ± 7.2 | 0.15 | 11.0 ± 16.7 | 10.8 ± 18.9 | 0.62 |
| LF/HF | 2.8 ± 2.4 | 2.9 ± 1.7 | 0.59 | 3.1 ± 1.7 | 2.4 ± 1.3 | 0.32 |
| LF/HF-SD | 3.2 ± 2.7 | 1.7 ± 1.8 | 0.086 | 2.2 ± 1.9* | 2.5 ± 2.5 | 0.77 |

[†]P-value: Compared between the DE and non-DE groups.

*Wilcoxon test: $P < 0.05$ compared pre- and post-ophthalmic solution administration.

(DE, dry eye; LF, low-frequency component; HF, high-frequency component, LF/HF, ratio of LF to HF; SD, standard deviation).

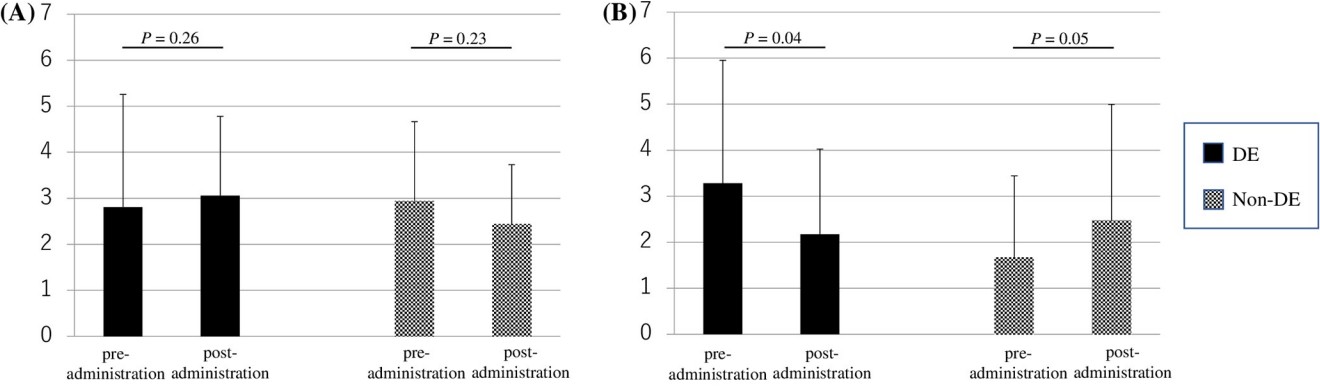

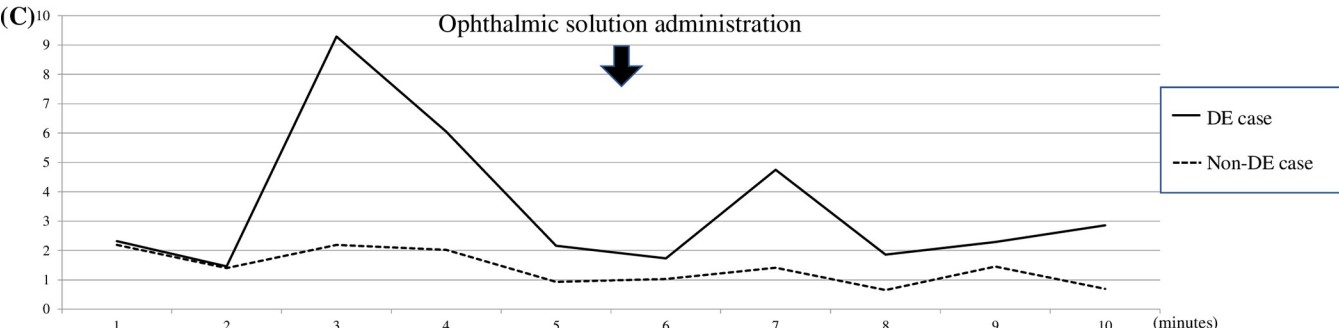

**Fig 2. The effect of instilling the ophthalmic solution on the LF/HF ratio and typical cases of fluctuations in the LF/HF ratio over time in the DE and non-DE groups.** (A) The LF/HF ratio pre- and post-instillation of ophthalmic solution in the DE and non-DE groups. There are no significant changes in either the DE or non-DE groups (P > 0.05). (B) The standard deviation of the LF/HF ratio pre- and post-instillation of ophthalmic solution in the DE and non-DE groups. The standard deviation of the LF/HF ratio significantly decreased post-instillation in the DE group (P < 0.05), whereas it was low pre-instillation and increased post-instillation in the non-DE group. (C) Typical DE and non-DE cases show fluctuation of the LF/HF ratio over time. The DE case presented is of a 39-year-old woman with a J-OSDI score of 18.8 points. The non-DE case is of a 21-year-old woman with a J-OSDI score of 9.1 points. The fluctuation in the LF/HF ratio of the DE case is larger than of the non-DE case, and it decreases post-instillation. DE, dry eye; HF, high-frequency component; J-OSDI, Japanese version of the Ocular Surface Disease Index Questionnaire; LF, low-frequency component.

keratoconjunctival scores, and LF/HF-SD. The LF/HF-SD value was a significant factor affecting the J-OSDI scores ($P < 0.05$), whereas the other variables were not.

## Discussion

Heart rate variability analysis has been widely used to assess autonomic nervous function [25,29–35]. The HF is believed to reflect cardiac parasympathetic nerve activity, while LF

**Table 3. Multiple regression analysis for determining factors affecting the J-OSDI scores.**

| Factor | Standardization coefficient (β) | t-value | P-value |
|---|---|---|---|
| (constant) | | 2.837 | 0.008 |
| Pre-ophthalmic solution administration LH/FH-SD | 0.469 | 3.015 | 0.005* |
| TBUT | −0.215 | −1.095 | 0.283 |
| Keratoconjunctival staining scores | 0.132 | −1.808 | 0.426 |
| Schirmer test values | −0.103 | −0.546 | 0.589 |

Dependent variable: J-OSDI scores, Adjusted R: 0.52.

*$P < 0.05$.

LF, low-frequency component; HF, high-frequency component; LF/HF, ratio of LF to HF; SD, standard deviation; TBUT, tear break-up time; J-OSDI, Japanese version of the Ocular Surface Disease Index.

mainly reflects sympathetic activity [29–31,33]. The LF/HF ratio has gained wide acceptance as a tool to assess cardiovascular autonomic regulation, where increases in the LF/HF ratio are assumed to reflect a shift to "sympathetic dominance" and decreases in this index to correspond to a "parasympathetic dominance" [31]. Positive and negative emotional states result in differential heart rate variability responses [36–40].

Based on the hypothesis that the symptoms of short TBUT DE may be induced by the central nervous system, which is associated with the autonomic nervous system, we measured autonomic nervous activity and investigated 1) whether short TBUT DE has specific characteristics with regards to autonomic activity and 2) if it is associated with clinical symptoms.

Short TBUT DE was characterized by substantial changes in autonomic balance over time. In addition, DE symptom intensity was related to fluctuations in autonomic balance over time. Accordingly, the onset of DE symptoms may be related to fluctuations in the autonomic balance. We initially expected sympathetic nerve activity to be high under the stress induced by unpleasant emotions caused by eye discomfort and/or pain in DE. However, the subjective stress levels observed in the DE group were not high and were similar to those in the non-DE group. There was no difference in the LF/HF ratio between the DE and non-DE groups, indicating that DE symptoms were not linked to either subjective stress or the magnitude of autonomic nervous activity. Instead, the LF/HF ratio fluctuations over time were larger in the DE group than those in the non-DE group, and DE symptom intensity was related to fluctuations in the LF/HF ratio, while DE parameters such as keratoconjunctival epithelial damage, tear stability, and amount of tear secretion, were not. The result that there was no difference in temperature sensation to a cold stimulus (instillation of eye drops) with or without DE indicates that peripheral nerve sensitivity is not involved in the induction of DE symptoms. It is speculated that changes in the central nervous system, rather than changes in the peripheral nervous system, are involved.

Fluctuations in autonomic activity can play an important role in disease pathogenesis. Living organisms are constantly subjected to changes and stimuli in their external environment and are equipped with mechanisms to maintain optimal functioning through regulating tissues and organs. The regulation of the autonomic nervous system is one such mechanism. However, DE, as a pathological condition, might cause the autonomic nervous system to initiate unstable physiological functioning. Fluctuations in autonomic activity are involved in the etiology of several diseases [41,42]. For example, a comparative study of carotid endarterectomy and carotid artery stenting, which are both treatments for hypertrophic myocardial infarction, showed that long-term autonomic fluctuations due to parasympathetic muscle tone hypertonia occur perioperatively to carotid endarterectomy, and increase the incidence of myocardial infarction [41]. A study of autonomic activity in patients with Brugada syndrome, a heart disease that causes idiopathic arrhythmias (ventricular fibrillation), showed that the circadian variation of the LF and LF/HF ratio were low in patients with the syndrome and that the LF/HF-SD and LF could be useful parameters for the diagnosis of Brugada syndrome [25]. Similarly, our study shows that such fluctuations may be associated with DE symptoms.

Fluctuations in the LH/HF ratio decreased post-ophthalmic solution administration in the DE group, but not in the non-DE group. However, there was no significant difference between the two groups in terms of the comfort level experienced by participants when they were exposed to the ophthalmic solution, suggesting that the LF/HF-SD was unrelated to comfort when exposed to a stimulus. These findings are contradictory. Temporarily improving DE symptoms by ophthalmic solution administration and feeling comfortable with ophthalmic solution administration may not be related.

Our study has a few limitations. First, we did not confirm if the participants had any autonomic dysfunctions other than DE. Menopausal disorders, depression, and some other

diseases are known to impair the balance of the autonomic nervous system in a broad sense; this could have been confirmed by performing an autonomic nerve test, such as a head-up tilt test or assessing changes in blood pressure due to changes in posture. Strictly speaking, it cannot be ruled out that these factors may have influenced the current results. However, since mental stress was similar between the DE and non-DE groups, it is unlikely that mental stress affected changes in autonomic nerve activity. Second, participants with decreased TBUT without DE symptoms were included in the non-DE group. Patients with decreased TBUT included those who remained asymptomatic (non-DE) and those who developed symptoms (DE). Thus, in this study, we compared the DE and non-DE groups, focusing on the presence or absence of symptoms. In fact, in Asia, DE diagnosis is defined as a case in which tear film instability and DE symptoms are present [5]. However, tear film instability without DE symptoms is not considered as confirmed DE diagnosis. Third, the sex of the participants in this study did not match between the DE and non-DE groups, possibly influencing the results. In addition, lifestyle habits such as smoking [42] and sleeping [43], which can be confounding factors and may affect the results, were not evaluated. In addition, since the autonomic nervous system is affected by environmental factors (e.g., temperature, humidity, and room illuminance) and/or body conditions (e.g., aging, drowsiness, and fatigue), it was difficult to assess autonomic nerve activity that reflected only DE.

## Conclusion

In conclusion, the fluctuation in the autonomic balance was large and related to DE symptom intensity in short TBUT DE. This fluctuation was decreased by instilling hyaluronic acid ophthalmic solution. Fluctuations in autonomic activity may be an important key to understanding the mechanism of DE symptoms.

## Supporting information

**S1 Table. Raw data.**
(PDF)

## Acknowledgments

### Ethics

The study protocol was reviewed and approved by the Ethics Committee of the Institutional Review Board of Ito Clinic, Saitama, Japan (registration number IRIN2021-11). All procedures were performed according to the ethical standards of the responsible committee on human experimentation (institutional and national) and the Helsinki Declaration of 1964, as revised in 2013. Written informed consent was obtained from all participants. This study was registered in the University Hospital Medical Information Network (registration number UMIN000045019).

## Author Contributions

**Conceptualization:** Minako Kaido.

**Data curation:** Minako Kaido, Reiko Ishida.

**Formal analysis:** Minako Kaido.

**Investigation:** Minako Kaido.

**Methodology:** Minako Kaido.

**Project administration:** Minako Kaido, Reiko Arita.

**Resources:** Minako Kaido.

**Software:** Minako Kaido.

**Supervision:** Reiko Arita, Yasue Mitsukura, Kazuo Tsubota.

**Validation:** Minako Kaido, Yasue Mitsukura.

**Visualization:** Minako Kaido.

**Writing – original draft:** Minako Kaido.

**Writing – review & editing:** Reiko Arita, Yasue Mitsukura, Kazuo Tsubota.

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
