## [Decision Letter · Decision Letter 0]

7 Sep 2022

PONE-D-22-23078Autonomic nerve activity in dry eye with decreased tear stabilityPLOS ONE

Dear Dr. Kaido,

Thank you for submitting your manuscript to PLOS ONE. After careful consideration, we feel that it has merit but does not fully meet PLOS ONE’s publication criteria as it currently stands. Therefore, we invite you to submit a revised version of the manuscript that addresses the points raised during the review process.

We look forward to receiving your revised manuscript.

Kind regards,

Munetaka Hirose, M.D., Ph.D.

Academic Editor

PLOS ONE

Reviewers' comments:

Reviewer's Responses to Questions

**Comments to the Author**

1. Is the manuscript technically sound, and do the data support the conclusions?

Reviewer #1: Yes

Reviewer #2: Partly

2. Has the statistical analysis been performed appropriately and rigorously? 

Reviewer #1: Yes

Reviewer #2: Yes

3. Have the authors made all data underlying the findings in their manuscript fully available?

Reviewer #1: Yes

Reviewer #2: Yes

4. Is the manuscript presented in an intelligible fashion and written in standard English?

Reviewer #1: Yes

Reviewer #2: Yes

5. Review Comments to the Author

Reviewer #1: General comments: The paper has some usefulness but it is requires careful editing, attention to detail and reorganisation to make it potentially publishable. This is a naïve report.

Abstract

The abstract has a poor structure. In many places there appear to be over-elaboration of findings into a story. I will encourage the authors to tidy and re-write the abstract.

I would urge the authors to briefly talk about how autonomic nerves in the ocular surface uses neuromodulators to maintain ocular surface homeostasis.

Kindly see these references

1. Hwang DD, Lee SJ, Kim JH, Lee SM. The role of neuropeptides in pathogenesis of dry dye. Journal of Clinical Medicine. 2021 Sep 19;10(18):4248.

2. Asiedu K, Markoulli M, Bonini S, Bron AJ, Dogru M, Kwai N, Poynten A, Willcox MD, Krishnan AV. Tear film and ocular surface neuropeptides: Characteristics, synthesis, signaling and implications for ocular surface and systemic diseases. Experimental eye research.:108973.

3. Sabatino F, Di Zazzo A, De Simone L, Bonini S. The intriguing role of neuropeptides at the ocular surface. The ocular surface. 2017 Jan 1;15(1):2-14.

Line 62-66 “Several studies on corneal sensory nerves related to neuropathic pain have focused on transient receptor potential vanilloid 1 (TRPV1), which is associated with pain sensation, and transient receptor potential melastatin 8 (TRPM8), which is associated with temperature change sensation on the corneal surface.” Kindly cite the following reference “Asiedu K. Role of ocular surface neurobiology in neuronal-mediated inflammation in dry eye disease. Neuropeptides.;95:102266.”

Line 49-60- There appears to be focus on the Japanese concept of dry eye. But Plos one is an international journal not a Japanese journal hence I will expect the authors to focus on the DEWS II report recommendations.

“Additionally, higher brain functions may be involved in the onset and progression of DE symptoms.” Provide references.

Include a sample size calculation and cursory look at your paper shows you may be under-powered.

In the statistical analysis section include normality testing statement if this was done.

“Only the eye with the more prominent symptoms was selected for evaluation in each 114 participant. When symptoms were equal in both eyes, the right eye was selected.” This does not make any sense and please kindly delete from the paper.

Discussion

From the numbering in your discussion.

“Based on the hypothesis that the symptoms of short TBUT DE may be induced by the 304 central nervous system, which is associated with the autonomic nervous system, we measured autonomic nervous activity and investigated 1) whether short TBUT DE has specific 15 characteristics with regards to autonomic activity and 2) if it is associated with clinical symptoms. In the present study, there was no difference in temperature sensation to a cold stimulus (drop of ophthalmic solution) with or without DE, demonstrating that peripheral 309 nerve sensitivity is not involved in the induction of DE symptoms”

Focus on the key findings and then later the less impactful findings. There are many places in this paper that requires citation.

Reviewer #2: 1. The title says automatic nerve activity in dry eye. However there was no difference in autonomic nerve activity represented as LF/HF ratio but the SD value of LF/HF ratio was different. So it seems necessary to modify the title to the variability of automatic nerve activity.

2. Ths study tried to investigate the relationship between the symptom and automatic balance fluorescence of Short TBUT DE, but it seems that the exclusion criteria for taking drugs or systemic diseases in the patient group should be clarified in order to properly analyze the study. No relevant information has been suggested in this study.

3. When classifying dry eye group and non-dry eye group, they were classified as symptoms, and TBUT did not meet the classification criteria. However, since this study targeted the Short TBUT group, it seems that classification as TBUT is also necessary.

6. PLOS authors have the option to publish the peer review history of their article (what does this mean?). If published, this will include your full peer review and any attached files.

Reviewer #1: No

Reviewer #2: No

---

## [Author Response · Author response to Decision Letter 0]

28 Sep 2022

We thank you for giving us the opportunity to revise our manuscript entitled “Variability of autonomic nerve activity in dry eye with decreased tear stability.” The manuscript ID is PONE-D-22-23078.

Reviewer #1: General comments: The paper has some usefulness but it is requires careful editing, attention to detail and reorganisation to make it potentially publishable. This is a naïve report.

We thank you for the time and effort spent throughout the review process of this manuscript. We believe that the excellent comments did increase the utility of this paper. We sincerely thank and send our gratitude to all parties involved throughout this revision process.

Abstract

The abstract has a poor structure. In many places there appear to be over-elaboration of findings into a story. I will encourage the authors to tidy and re-write the abstract.

I would urge the authors to briefly talk about how autonomic nerves in the ocular surface uses neuromodulators to maintain ocular surface homeostasis.

Kindly see these references

1. Hwang DD, Lee SJ, Kim JH, Lee SM. The role of neuropeptides in pathogenesis of dry dye. Journal of Clinical Medicine. 2021 Sep 19;10(18):4248.

2. Asiedu K, Markoulli M, Bonini S, Bron AJ, Dogru M, Kwai N, Poynten A, Willcox MD, Krishnan AV. Tear film and ocular surface neuropeptides: Characteristics, synthesis, signaling and implications for ocular surface and systemic diseases. Experimental eye research.:108973.

3. Sabatino F, Di Zazzo A, De Simone L, Bonini S. The intriguing role of neuropeptides at the ocular surface. The ocular surface. 2017 Jan 1;15(1):2-14.

Please note that we clearly stated the purpose of the study by clarifying the relationship between neurogenic inflammation and autonomic nerve activity due to decreased tear film stability, and briefly summarized the entire abstract on line 30-35 as follows: “Neurological abnormalities in the pathogenesis of dry eye (DE) have been highlighted. The autonomic nervous system plays a crucial role in the maintenance of homeostasis. Neurogenic inflammation due to decreased stability of tear film may induce changes in autonomic nerve activity, involving in symptom expression.This study aimed to measure biological parameters that represent autonomic nerve activity in DEs caused by tear film dysfunction and investigate their relationship with symptom intensity.”

Along with the change in the abstract, the introduction has also been changed.

Line 62-66 “Several studies on corneal sensory nerves related to neuropathic pain have focused on transient receptor potential vanilloid 1 (TRPV1), which is associated with pain sensation, and transient receptor potential melastatin 8 (TRPM8), which is associated with temperature change sensation on the corneal surface.” Kindly cite the following reference “Asiedu K. Role of ocular surface neurobiology in neuronal-mediated inflammation in dry eye disease. Neuropeptides.;95:102266.”

We cited the paper pointed out.

Line 49-60- There appears to be focus on the Japanese concept of dry eye. But Plos one is an international journal not a Japanese journal hence I will expect the authors to focus on the DEWS II report recommendations.

We rewrote as follows on line 52-82: “There is a discrepancy between the symptoms and clinical findings of dry eye (DE) [1,2]. DE is classified into aqueous-deficient DE and evaporative DE. However, the concept that decreased tear film stability is of essence to DE is gaining prominence, especially in Asia [3].”

We also added the information in the next paragraph in line 92-95 as follows: “The Dry Eye Workshop II identified neuropathy as one of the etiologies of DE disease [8]. Neuropathic pain due to neurogenic inflammation caused by damage or irritation to sensory nerve endings on the ocular surface has been implicated as an important cause of DE symptoms [9-12].”

“Additionally, higher brain functions may be involved in the onset and progression of DE symptoms.” Provide references.

Please note that the references were cited.

Yu K, Guo Y, Ge QM, Su T, Shi WQ, Zhang LJ, et al. Altered spontaneous activity in the frontal gyrus in dry eye: a resting-state functional MRI study. Sci Rep. 2021;11(1):12943. doi: 10.3389/fncel.2020.598678

Sun T, Shu HY, Wu JL, Su T, Liu YJ, Zhang LJ, et al. Investigation of changes in the activity and function of dry eye-associated brain regions using the amplitude of low-frequency fluctuations method. Biosci Rep. 2022;42(1):BSR20210941. doi:10.1042/BSR20210941

Liang RB, Liu LQ, Shi WQ, Sun T, Ge QM, Li QY, et al. Abnormal fractional amplitude of low frequency fluctuation changes in patients with dry eye disease: a functional magnetic resonance imaging study. Front Hum Neurosci. 2022;16:900409. doi:10.3389/fnhum.2022.900409

Include a sample size calculation and cursory look at your paper shows you may be under-powered.

We clarified how the sample size was determined in the section of Statistical analyses on line 272-275 as follow: “The sample size was determined based on a previous study [32]. With a mean difference of 0.51 between the two groups, a standard deviation of 0.5, an alpha error of 0.05, and a power of 0.8, the required sample size was 16 in each group. Forty cases were registered for consideration of ineligible cases.”

In the statistical analysis section include normality testing statement if this was done.

We had performed the t-test without a normality test. As you kindly and precisely pointed out, we performed the normality test, and found out that it was difficult to say that the distribution was normal. Thus, I re-analyzed it with the Mann-Whitney test. Along to re-analyzing, the data of the results have been changed, but there is no change in the outline of the results.

In the section of Statistical analyses, the following statement was added on line 276-287, such as that: " Baseline J-OSDI scores, tear function, and autonomic nerve activity were compared between both groups using the Mann-Whitney test, since it was difficult to regard the normal distribution as a result of the normality test. Heart rate variability parameters pre- and post-ophthalmic solution administration were compared in each group using the Wilcoxon test. The temperature sensation and the comfort level to the ophthalmic solution were also compared between both groups using the Mann-Whitney test."

According to the analysis method changed, the P value for each evaluation item changed.

“Only the eye with the more prominent symptoms was selected for evaluation in each 114 participant. When symptoms were equal in both eyes, the right eye was selected.” This does not make any sense and please kindly delete from the paper.

Please note that it was deleted.

Discussion

From the numbering in your discussion.

“Based on the hypothesis that the symptoms of short TBUT DE may be induced by the central nervous system, which is associated with the autonomic nervous system, we measured autonomic nervous activity and investigated 1) whether short TBUT DE has specific 15 characteristics with regards to autonomic activity and 2) if it is associated with clinical symptoms. In the present study, there was no difference in temperature sensation to a cold stimulus (drop of ophthalmic solution) with or without DE, demonstrating that peripheral nerve sensitivity is not involved in the induction of DE symptoms”

Focus on the key findings and then later the less impactful findings. There are many places in this paper that requires citation.

We deleted the sentence, "In the present study, there was no difference in temperature sensation to a cold stimulus…." in the second paragraph, and rewrote on line 485-488 as follows: “Moreover, the result that there was no difference in temperature sensation to a cold stimulus (instillation of eye drops) with or without DE indicates that peripheral nerve sensitivity is not involved in the induction of DE symptoms.”

Reviewer #2:

1. The title says automatic nerve activity in dry eye. However there was no difference in autonomic nerve activity represented as LF/HF ratio but the SD value of LF/HF ratio was different. So it seems necessary to modify the title to the variability of automatic nerve activity.

The title was changed to ”Variability of a Autonomic nerve activity in dry eye with decreased tear stability.”

2. Ths study tried to investigate the relationship between the symptom and automatic balance fluorescence of Short TBUT DE, but it seems that the exclusion criteria for taking drugs or systemic diseases in the patient group should be clarified in order to properly analyze the study. No relevant information has been suggested in this study.

Pleasse note that we restated about exclusion criteria in the section of “Participants” on line 175-180 as follows: “Exclusion criteria were previous ocular surgery and/or trauma within the last 12 months; anatomic abnormalities in the cornea, conjunctivitis, and/or eyelids; glaucoma; current contact lens use; and systemic diseases that may affect autonomic nerve activity such as heart disease excluding hypertension, diabetes mellitus, Parkinson's disease, pulmonary disease, and Sjögren’s syndrome.”

We also added the state in the result section of “Demographics of the study population” on line 303-305 as follows: “Regarding the presence or absence of systemic disease, hypertension was observed in 3 participants each in the DE and non-DE groups, and breast cancer in remission was seen in 1 particpant in the DE group.”

We also rewrote the limitation on line 516-522 as follows: “Menopausal disorders, depression, and some other diseases are known to impair the balance of the autonomic nervous system in a broad sense; this could have been confirmed by performing an autonomic nerve test, such as a head-up tilt test or assessing changes in blood pressure due to changes in posture. Strictly speaking, it cannot be ruled out that these factors may have influenced the current results. However, since mental stress was similar between the DE and non-DE groups, it is unlikely that mental stress affected changes in autonomic nerve activity.”

3. When classifying dry eye group and non-dry eye group, they were classified as symptoms, and TBUT did not meet the classification criteria. However, since this study targeted the Short TBUT group, it seems that classification as TBUT is also necessary.

It is pointed in our previous paper* that many cases only with decreased BUT but no DE symptoms exist. We are interested in the mechanism by which cases with decreased BUT develop DE symptoms. In this study, we compared between DE and non-DE, focusing on the presence or absence of symptoms. 

In addition, DE diagnostic criteria in Japan defines DE as a case in which BUT is 5 seconds or less and DE symptoms are observed.

Please note that we mentioned it in the limitation section on line 522-528 as follows: “Second, participants with decreased TBUT without DE symptoms were included in the non-DE group. Patients with decreased TBUT included those who remained asymptomatic (non-DE) and those who developed symptoms (DE). Thus, in this study, we compared the DE and non-DE groups, focusing on the presence or absence of symptoms. In fact, in Asia, DE diagnosis is defined as a case in which tear film instability and DE symptoms are present [3]. However, tear film instability without DE symptoms is not considered as confirmed DE diagnosis.”

*M Kaido et al. the Relation of accommodative microfluctuation with dry eye symptoms in short tear break-up time dry eye. PLoS One. 2017 8;12

---

## [Decision Letter · Decision Letter 1]

18 Oct 2022

Variability of autonomic nerve activity in dry eye with decreased tear stability

PONE-D-22-23078R1

Dear Dr. Kaido,

We’re pleased to inform you that your manuscript has been judged scientifically suitable for publication and will be formally accepted for publication once it meets all outstanding technical requirements.

Kind regards,

Blanka Golebiowski, PhD BOptom

Academic Editor

PLOS ONE

Additional Editor Comments (optional):

Reviewers' comments:

Reviewer's Responses to Questions

**Comments to the Author**

1. If the authors have adequately addressed your comments raised in a previous round of review and you feel that this manuscript is now acceptable for publication, you may indicate that here to bypass the “Comments to the Author” section, enter your conflict of interest statement in the “Confidential to Editor” section, and submit your "Accept" recommendation.

Reviewer #1: All comments have been addressed

Reviewer #2: All comments have been addressed

2. Is the manuscript technically sound, and do the data support the conclusions?

Reviewer #1: Partly

Reviewer #2: Yes

3. Has the statistical analysis been performed appropriately and rigorously? 

Reviewer #1: N/A

Reviewer #2: Yes

4. Have the authors made all data underlying the findings in their manuscript fully available?

Reviewer #1: Yes

Reviewer #2: Yes

5. Is the manuscript presented in an intelligible fashion and written in standard English?

Reviewer #1: Yes

Reviewer #2: Yes

6. Review Comments to the Author

Reviewer #1: Delete from the abstract.” Neurological abnormalities in the pathogenesis of dry eye (DE) have been highlighted”

“Neurogenic inflammation due to decreased stability of tear film may induce changes in 30 autonomic nerve activity, involving in symptom expression” Re-phrase

“DE is classified into aqueous-deficient DE and evaporative DE.” Add these references

1. Lemp MA, Crews LA, Bron AJ, Foulks GN, Sullivan BD. Distribution of aqueous-deficient and evaporative dry eye in a clinic-based patient cohort: a retrospective study. Cornea. 2012 May;31(5):472-8. doi: 10.1097/ICO.0b013e318225415a. PMID: 22378109.

2. Asiedu K, Dzasimatu SK, Kyei S. Clinical subtypes of dry eye in youthful clinical sample in Ghana. Cont Lens Anterior Eye. 2019 Apr;42(2):206-211. doi: 10.1016/j.clae.2018.10.005. Epub 2018 Oct 15. PMID: 30337142.

‘The autonomic nervous system plays a crucial role in the maintenance of homeostasis. 83 In the field of DE, the parasympathetic nervous system, which is activated by abdominal 84 breathing, is important to increases tear volume [21]” Correct the grammar error in the last part of the sentence.

“The sample size was determined based on a previous study [32]. With a mean difference 191 of 0.51 between the two groups, a standard deviation of 0.5, an alpha error of 0.05, and a 192 power of 0.8, the required sample size was 16 in each group.” This should be in a different subheading .

Tidy up you discussion a bit more.

Reviewer #2: Much of the paper has been revised and written without major errors. I want to give acceptance on their paper.

7. PLOS authors have the option to publish the peer review history of their article (what does this mean?). If published, this will include your full peer review and any attached files.

Reviewer #1: No

Reviewer #2: No

---

## [Editor Report · Acceptance letter]

26 Oct 2022

PONE-D-22-23078R1 

Variability of autonomic nerve activity in dry eye with decreased tear stability 

Dear Dr. Kaido:

I'm pleased to inform you that your manuscript has been deemed suitable for publication in PLOS ONE. Congratulations! Your manuscript is now with our production department. 

Kind regards, 

on behalf of

Associate Professor Blanka Golebiowski 

Academic Editor

PLOS ONE